# A Modified Meiotic Recombination in *Brassica napus* Largely Improves Its Breeding Efficiency

**DOI:** 10.3390/biology10080771

**Published:** 2021-08-13

**Authors:** Franz Boideau, Alexandre Pelé, Coleen Tanguy, Gwenn Trotoux, Frédérique Eber, Loeiz Maillet, Marie Gilet, Maryse Lodé-Taburel, Virginie Huteau, Jérôme Morice, Olivier Coriton, Cyril Falentin, Régine Delourme, Mathieu Rousseau-Gueutin, Anne-Marie Chèvre

**Affiliations:** 1IGEPP, INRAE, Institut Agro, Université de Rennes, 35650 Le Rheu, France; franz.boideau@inrae.fr (F.B.); alexandre.pele@outlook.fr (A.P.); coleen.tng@gmail.com (C.T.); gwenn.trotoux@inrae.fr (G.T.); freebe1952@gmail.com (F.E.); loeiz.maillet@inrae.fr (L.M.); marie-madeleine.gilet@inrae.fr (M.G.); maryse.taburel@inrae.fr (M.L.-T.); virginie.huteau@inrae.fr (V.H.); jerome.morice@inrae.fr (J.M.); olivier.coriton@inrae.fr (O.C.); cyril.falentin@inrae.fr (C.F.); regine.delourme@inrae.fr (R.D.); mathieu.rousseau-gueutin@inrae.fr (M.R.-G.); 2Laboratory of Genome Biology, Institute of Molecular Biology and Biotechnology, Adam Mickiewicz University in Poznan, 61-614 Poznan, Poland

**Keywords:** recombination rate and distribution, polyploidy, allotriploidy, *Brassica napus*, genetic mapping, plant breeding

## Abstract

**Simple Summary:**

The selection of varieties more resilient to disease and climate change requires generating new genetic diversity for breeding. The main mechanism for reshuffling genetic information is through the recombination of chromosomes during meiosis. We showed in oilseed rape (*Brassica napus*, AACC, 2*n* = 4*x* = 38), which is a natural hybrid formed from a cross between turnip (*B. rapa*, AA, 2*n* = 2*x* = 20) and cabbage (*B. oleracea*, CC, 2*n* = 2*x* = 18), that there is significantly more crossovers occurring along the entire A chromosomes in allotriploid AAC (crossbetween *B. napus* and *B. rapa*) than in diploid AA or allotetraploid AACC hybrids. We demonstrated that these allotriploid AAC hybrids are highly efficient to introduce new variability within oilseed rape varieties, notably by enabling the introduction of small genomic regions carrying genes controlling agronomically interesting traits.

**Abstract:**

Meiotic recombination is the main tool used by breeders to generate biodiversity, allowing genetic reshuffling at each generation. It enables the accumulation of favorable alleles while purging deleterious mutations. However, this mechanism is highly regulated with the formation of one to rarely more than three crossovers, which are not randomly distributed. In this study, we showed that it is possible to modify these controls in oilseed rape (*Brassica napus*, AACC, 2*n* = 4*x* = 38) and that it is linked to AAC allotriploidy and not to polyploidy *per se*. To that purpose, we compared the frequency and the distribution of crossovers along A chromosomes from hybrids carrying exactly the same A nucleotide sequence, but presenting three different ploidy levels: AA, AAC and AACC. Genetic maps established with 202 SNPs anchored on reference genomes revealed that the crossover rate is 3.6-fold higher in the AAC allotriploid hybrids compared to AA and AACC hybrids. Using a higher SNP density, we demonstrated that smaller and numerous introgressions of *B. rapa* were present in AAC hybrids compared to AACC allotetraploid hybrids, with 7.6 Mb vs. 16.9 Mb on average and 21 *B. rapa* regions per plant vs. nine regions, respectively. Therefore, this boost of recombination is highly efficient to reduce the size of QTL carried in cold regions of the oilseed rape genome, as exemplified here for a QTL conferring blackleg resistance.

## 1. Introduction

Meiotic recombination shuffles parental alleles to produce new allelic combinations in the progenies, hence producing new genetic diversity at each generation. This biological mechanism is a key evolutionary process that is commonly used in plant breeding to accumulate favorable alleles and purge deleterious mutations [1,2,3]. This phenomenon occurs during meiosis, a specialized eukaryotic cell division that gives rise to haploid gametes via a single round of DNA replication followed by two rounds of chromosome segregation [4]. The balanced segregation of homologs during the first meiotic division is ensured by meiotic recombination through crossovers that temporarily connect homologous chromosomes until metaphase I. For successful chromosome segregation and hence the production of viable gametes, such joined homolog pairs require at least one crossover. Besides this crucial role, crossovers result in reciprocal exchanges between homologous non-sister chromatids, generating new allelic combinations in gametes.

However, crossover formation is under tight genetic control [4]. Typically, eukaryotes experience one obligatory crossover per chromosome pair and per meiosis, but rarely more than three, despite a large number of generated DNA double strand breaks (DSBs). For instance, out of the 150 to 250 DSBs generated per meiosis, only ~10 result in crossovers in *Arabidopsis*, the others giving rise to non-crossovers (NCOs) [5,6]. In the last decade, several proteins were highlighted to promote the repair of DSBs into NCOs in *A. thaliana* (e.g., FANCM, RECQ4, FIGL1, HCR1), thereby limiting the overall number of crossovers generated per meiosis [7,8,9,10]. Furthermore, crossovers are unevenly distributed along chromosomes [4], with a gradient from the telomere to the centromere, as exemplified in bread wheat [11], maize [12] or potato [13]. Fine characterization of crossover distributions has pointed out links with genomic and epigenetic features, revealing that crossovers preferentially occur close to gene promoters and terminators in euchromatic regions that are hypomethylated and enriched in H3K4me3 histone marks [14,15,16,17]. On the contrary, they are greatly reduced or even prevented in regions that are heavily methylated and enriched in H3K9me2, such as pericentromeres [18,19,20], or regions exhibiting sequence variations (e.g., insertions, deletions, inversions, translocation, etc.) [21]. Consequently, most crossovers occur in a minor proportion of the chromosome, as exemplified in *A. thaliana*, for which 80% of crossovers formed are observed within less than 26% of the genome [14].

Despite the strong regulation of meiotic recombination, extensive variations are observed for crossover number and distribution both between and within species [22,23]. Factors responsible for these modifications have the potential to profoundly influence selective responses and adaptation while accelerating plant breeding programs [24,25,26,27]. In a non-exhaustive list of examples, environmental conditions (e.g., abiotic stress, temperature) (for a review, see [28]), sex of meiosis [29,30], genetic background [31,32,33] and ploidy level (for a review, see [34]) have been linked to variations in crossover level and/or pattern. The most striking in terms of crossover reshuffling and promising in terms of breeding applications is the last factor listed, with the particular case of *Brassica* AAC allotriploids (2*n* = 3*x* = 29). Indeed, in these viable and fertile hybrids, resulting from the cross between the rapeseed *B. napus* (AACC, 2*n* = 4*x* = 38) and its *B. rapa* diploid progenitor (AA, 2*n* = 2*x* = 20), crossover frequency obtains an unprecedent boost on the A07 chromosome when compared to the results obtained from diploid or allotetraploid hybrids [35]. The A chromosomes show an average recombination frequency that is 3.4-fold higher compared to diploid AA hybrids, carrying identical A genomes, when AAC plants are used as female parents [36]. Moreover, the authors pointed out that this crossover boost is strikingly associated with dramatic changes in the shape of recombination landscapes. Indeed, AAC allotriploids frequently exhibit crossovers in genomic regions that are normally totally deprived of any recombination event in AA diploids, such as the pericentromeres. The molecular mechanisms responsible for this unique recombination landscape observed in *Brassica* AAC allotriploids are currently being investigated but are yet to be deciphered. Nevertheless, it is feasible to recover an AACC genomic structure from AAC plants [37], enabling us to broaden the oilseed rape genetic diversity that has been severely eroded in recent decades due to a high selective pressure applied for yield and seed quality traits [38]. Interspecific crosses between *B. napus* and its *B. rapa* diploid progenitor, combined with reshaped homologous recombination, would facilitate small introgressions of targeted valuable loci deriving from *B. rapa* while preserving the agronomic value of *B. napus* cultivars. Recently, simulations conducted with AA and AAC *Brassica* genetic maps tend to confirm this putative benefit [39,40]. The authors demonstrated the genetic gain resulting from the use of allotriploids in long-term selection programs, at least in non/low-recombining pericentromeric regions. So far, these analyses have been limited to the comparison between AA and AAC hybrids as no comparable genetic maps have been generated to date for allotetraploid AACC hybrids with the same A genetic background as in AAC hybrids. This prevents addressing the role played by polyploidy *per se* in this recombination pattern, as well as the true potential of using allotriploids compared to allotetraploids in oilseed rape breeding programs. Moreover, the potential of using AAC allotriploids (compared to allotetraploids) in reducing the size of a quantitative trait loci (QTL) identified in *B. napus* remains to be assessed.

In the present study, we analyzed how this unique recombination pattern observed in *Brassica* AAC allotriploid hybrids can be used for oilseed rape breeding. Firstly, we tested the impact of polyploidy *per se* vs. AAC allotriploidy by comparing the recombination profile in AA, AAC and AACC hybrids carrying an identical A genotype. We confirmed that the modification of meiosis control only occurs in AAC allotriploid hybrids. Secondly, compared to a previous study [36], we densified the genetic maps from the AAC or AACC hybrids (10 times more SNPs) and finely compared the size and distribution of *B. rapa* introgressions observed along each A chromosome, depending on the hybrid ploidy level. Finally, we highlighted the interest of using allotriploids in breeding programs, by taking the example of a large resistance QTL against *Leptosphaeria maculans* [41] present in a cold recombining pericentromeric region.

## 2. Materials and Methods

### 2.1. Plant Material

We compared crossover rate between homologous A chromosomes of *Brassica* hybrids presenting three different ploidy levels but sharing a genetically identical A genome sequence (Figure 1). To that purpose, we used the plant material already described by Pelé et al. [36]: (1) the diploid AnAr (2*n* = 2*x* = 20) F1 hybrid, which was obtained by crossing the diploid AnAn (2*n* = 2*x* = 20) genotype extracted from the French variety *B. napus* cv. Darmor AnAnCnCn (2*n* = 4*x* = 38) [42] with the pure inbred line *B. rapa* ssp. *pekinensis* cv. Chiifu-401-42 (male, ArAr, 2*n* = 2*x* = 20), was backcrossed with the Korean variety *B. napus* cv. Yudal and gave rise to 329 plants; (2) the allotriploid hybrid AnArCn (2*n* = 3*x* = 29), which was obtained by crossing the allotetraploid B*. napus* cv. Darmor AnAnCnCn (2*n* = 4*x* = 38) and the pure inbred line *B. rapa* ssp. *pekinensis* cv. Chiifu (male, ArAr, 2*n* = 2*x* = 20), was then backcrossed with the Korean variety *B. napus* cv. Yudal and gave rise to 109 plants. This plant material was complemented by backcrossing the AnArCn hybrid (2*n* = 3*x* = 29) to *B. napus* cv. Darmor (131 plants selected among 234). To produce the allotetraploid hybrid AnArCnCo (2*n* = 4*x* = 38), we first crossed the pure inbred line *B. rapa* cv. Chiifu (ArAr, 2*n* = 2*x* = 20) with the doubled haploid *B. oleracea* cv. HDEM (CoCo, 2*n* = 2*x* = 18) and then performed embryo rescue on the obtained amphihaploid (ArCo, 2*n* = 19) as described in [43]. This hybrid spontaneously doubled its genomes and gave rise to the resynthesized allotetraploid ArArCoCo (2*n* = 4*x* = 38), hereafter referred as ChEM. This latter plant was crossed with *B. napus* cv. Darmor as female, and the AnArCnCo F1 hybrid gave rise to 213 plants after backcrossing with *B. napus* cv. Darmor as male. All parental accessions were provided by the Biological Resource Center BrACySol (Ploudaniel, France).

### 2.2. Flow Cytometry and Cytogenetic Studies

Chromosome numbers were assessed in leaves by flow cytometry as described by [37]. For the establishment of meiotic behavior, samples of young floral buds were fixed in Carnoy’s solution (alcohol:chloroform:acetic acid, 6:3:1) for 24 h at room temperature and stored until use in 50% ethanol at 4 °C. Anthers were then squashed and stained with 1% aceto-carmine. Chromosome pairing was assessed per plant from 20 pollen mother cells (PMCs) at metaphase I. For analyzing pairing between A and C chromosomes at metaphase I, BAC FISH was performed as described in [37] using the *B. oleracea* BoB014O06 BAC [44] as “genomic in situ hybridization-like” (GISH-like) to specifically distinguish all C chromosomes.

### 2.3. DNA Extraction and SNP Genotyping

Genomic DNA was extracted from lyophilized young leaves with the sbeadex maxi plant kit (LGC Genomics, Teddington Middlesex, UK) on the oKtopure robot at the GENTYANE platform (INRAE, Clermont-Ferrand, France). Genotyping data were obtained using the 202 SNP markers already defined [36], and revealed by Biomark^TM^ HD system (Fluidigm technology) and KASPar^TM^ chemistry (GENTYANE platform INRAE, Clermont-Ferrand, France), as well as from the *Brassica* 15K Illumina Infinium SNP array (SGS-TraitGenetics GmbH, Gatersleben, Germany). The context sequences of each SNP marker were physically localized on the reference genome *B. rapa* Chiifu v1.5 [45] and *B. napus* cv. Darmor-bzh v10 [46] using BLASTn (ver. 2.9.0, min. e-value 1 × 10^−20^) [47] and by keeping the best blast hit obtained for a given subgenome (minimum percentage of alignment and identity: 80%). SNPs that were polymorphic between the parental genotypes (i.e., AA in *B. rapa* and BB in *B. napus,* or vice versa*)* were selected for either the establishment of genetic maps or assessment of introgressed regions from the diploid parent into the *B. napus* genome.

### 2.4. Genetic Maps

The first genetic maps were established separately for the AnAr, the AnArCn and the AnArCnCo populations using the CarthaGene software (v. 1.2.3, [48]). Establishment of linkage groups and SNP ordering were examined using a logarithm of odds score (LOD) threshold of 4.0 and a maximum recombination frequency of 0.4, as described in [36]. Potential double crossover supported by only one genetic marker and with a physical distance between these two events below 500 kb was corrected as missing data, as described in [21]. After these few corrections, the final genetic maps were created using the Kosambi function to evaluate the genetic distances in centimorgans (cM) between linked SNP markers [49].

### 2.5. Characterization of the B. rapa Introgressions

Introgressions were determined by two methods, if either one SNP or at least two consecutive SNP markers were detected as heterozygous in each plant from the backcross progeny. To estimate the length of the introgresssed regions detected by only one SNP and to prevent the underestimation of the smallest introgressions, we considered the position of the consecutive previous and following SNPs. The positions of the first and last heterozygous SNPs were used to infer the size of introgressed regions defined by at least two consecutive SNPs, preventing the overestimation of introgressions. Each SNP can be present in introgressions of different sizes in different plants of a progeny and we calculated per population for each SNP position the average size of the introgressions carrying this marker.

### 2.6. Localization of a QTL of Interest

A quantitative trait locus (QTL) involved in blackleg (*Leptosphaeria maculans*) resistance [41] and present on the *B. napus* cv. Darmor A01 chromosome was selected for its pericentromeric localization. The context sequences of the SNP markers (derived from *Brassica* 60 K Illumina infinium array) flanking this QTL were retrieved and physically localized on the *B. napus* reference genome Darmor-bzh v10 [46] using BLASTn [47].

### 2.7. Inferring the Position of Centromeric and Pericentromeric Regions

The centromeric regions were defined by blasting several centromeric-specific repeats (CentBr1, CentBr2, TR238, TR805, PCRBr and CRB; [50,51]) against *B. napus* cv. Darmor-bzh v10 [46] and by examining the plot density of the BLASTn results (e-value less than 1 × 10^−20^). The pericentromeric borders for each chromosome were inferred by using the mean gene density along the chromosomes (as presented in [52]) and defined as the regions surrounding the centromere with a gene density below the chromosome average.

### 2.8. Statistical Analyses

The heterogeneity of crossover rates among progenies was assessed for every interval between consecutive SNP markers using a 2-by-2 chi-squared analysis considering a significance threshold of 5%. Additionally, the heterogeneity of crossover rates among progenies was evaluated at chromosome and genome scales using 2-by-2 chi-squared tests. For these tests, a conservative Bonferroni-corrected threshold of 5% was applied, using the number of intervals between adjacent SNP markers per A chromosome or for the A genome (as described in [36]).

The relationships between the relative size of introgressions normalized per A chromosome (%) vs. their relative distance from the centromeres (%) were studied by regression analyses, using the Spearman rank correlation, for linear (y = ax + b) and order 2 polynomial (y = ax² + bx + c) regressions (as described in [36]).

## 3. Results

### 3.1. Impact of Brassica Hybrid Genomic Structure (Diploid, Allotriploid or Allotetraploid) on Homologous Recombination Frequency and Distribution

To accurately compare the homologous recombination profile between hybrids of different ploidy levels in *Brassica*, we generated AnAr, AnArCn and AnArCnCo F1 hybrids that present the same A genome nucleotidic sequence originating from *B. napus* cv. Darmor (AnAnCnCn) and *B. rapa* cv. Chiifu (ArAr) (Figure 1). The diploid AnAr and the allotriploid AnArCn F1 hybrids have been described by [36]. Here, we also studied the meiotic behavior of different allotriploid AnArCn hybrids (Appendix A) and observed that they generally showed the expected meiotic configuration with 9 C univalents and 10 A bivalents as revealed by GISH-like analyses (Figure 2a). To produce the backcrossed progeny, we selected an AAC hybrid showing more than 80% of pollen mother cells with the expected configurations. In the backcross progeny of this hybrid, we were able to retrieve two plants with an AACC genomic structure over the 234 plants generated. Moreover, to create the allotetraploid AnArCnCo hybrid, we first produced the resynthesized *B. napus* ChEM by crossing *B. rapa* cv. Chiifu with *B. oleracea* cv. HDEM, followed by genome doubling of the resulting AC amphihaploid hybrid. Meiotic stability of the resynthesized allotetraploid was thereafter studied, revealing a relatively regular meiotic behavior with 70% of pollen mother cells in metaphase I exhibiting 19 bivalents (Appendix A). These results were in agreement with GISH-like analyses, which revealed a regular pairing (Figure 2b) with rare A and C chromosome pairing (Figure 2c). After crossing the resynthesized *B. napus* ChEM to *B. napus* cv. Darmor, we selected the most stable F1 hybrid (Appendix A, Figure 2d) and produced 213 progeny plants via a backcross to *B. napus* cv. Darmor. This progeny was genotyped with the same 202 SNPs used in [36], allowing a reliable comparison of the genetic maps derived from AnAr, AnArCn and AnArCnCo hybrids.

In total, cumulated genetic maps of the 10 A chromosomes of AnAr, AnArCn and AnArCnCo hybrids showed 830.2, 2902.3 and 790.8 cM, respectively. Pairwise comparison indicated a similar level of recombination at the whole A genome scale between the AnAr and AnArCnCo hybrids (Bonferroni-corrected chi-square test, *p* = 0.18). However, significant variations were detected for 3*x* vs. 2*x* and 3*x* vs. 4*x*. In both cases, A genome-wide crossover rate was on average 3.6-fold higher in the allotriploid hybrids (Bonferroni-corrected chi-square test, *p* < 2.2 × 10^−16^). After anchoring the polymorphic SNPs on the *B. rapa* cv. Chiifu v1.5 Ar genome, crossover distribution was analyzed by assessing heterogeneity of crossover rates among progenies for every interval between adjacent SNP markers. Similar recombination landscapes were thus identified between the AnAr and AnArCnCo hybrids as they evidenced significant differences in crossover rate in only two intervals out of 192 (1%), spanning 3.19 and 1.53 Mb on chromosomes A02 and A06, respectively (Bonferroni-corrected chi-square test, *p* < 0.05). However, comparison of AnArCn with AnAr and AnArCnCo hybrids revealed significant differences in crossover rate in more than 60% (133 and 118 over 192, respectively) of the intervals (*p* < 0.05) (Figure 3). Importantly, from the 20 intervals surrounding the centromeric regions (1 Mb from each centromeric border) of the 10 A chromosomes, we noticed that the recombination frequencies were always higher in the allotriploids compared to the diploid and allotetraploid hybrids. Significant differences were detected for 18 and 10 intervals in the AnArCn vs. AnAr and AnArCn vs. AnArCnCo comparisons, respectively (*p* < 0.05). Therefore, we concluded that recombination landscapes along A chromosomes of the allotriploid hybrid differed in a similar way, in regard to pericentromeric regions, to those of the diploid and allotetraploid hybrids.

### 3.2. Impact of Recombination on the Size and Distribution of Introgression within Oilseed Rape

To obtain a finer analysis of the crossover distribution in the allotriploid and allotetraploid hybrids, we then analyzed their progenies using the 15K SNP Illumina infinium array. A total of 2340 polymorphic SNP markers were physically anchored on the *B. napus* cv. Darmor-bzh v10 A genome, increasing the number of SNPs used for the comparison by more than ten times. On average, a polymorphic marker was observed every 150 kb along each A chromosome. Using this larger number of markers, we obtained genetic maps of 3045.4 and 827.7 cM for AnArCn and AnArCnCo hybrids, respectively. We thus improved the accuracy of our comparison and confirmed a higher recombination frequency in AnArCn (3.7-fold) compared to AnArCnCo hybrids (*p* < 2.2 × 10^−16^).

Given this extraordinary reshuffling of meiotic recombination occurring in allotriploids, we analyzed the size and distribution of Ar introgressions within the An oilseed rape genome according to the genomic structure of the F1 hybrids. The overall introgressed *B. rapa* genetic diversity introduced per generation is similar in allotriploid and allotetraploid hybrids (50% and 48%). However, as expected, with a larger number of crossovers occurring per chromosome pair and per meiosis, we observed that the mean introgression sizes were significantly smaller when arising from the meiosis of allotriploid compared to allotetraploid hybrids, with an average of 7.6 Mb vs. 16.9 Mb, respectively (Mann–Whitney–Wilcoxon test, *p* = 6.4 × 10^−104^) (Figure 4 and Figure 5a). As a second effect of crossover boost, significantly more introgressions from *B. rapa* were detected in backcross progeny of allotriploid hybrids with on average 21 heterozygous regions (on the 10 A chromosomes) per plant vs. nine regions in the backcross progeny of the allotetraploid hybrid (Mann–Whitney–Wilcoxon test, *p* = 2.3 × 10^−53^) (Figure 5b). Interestingly, the size of introgressions varied along the chromosome arms. The largest introgressions tended to colocalize around pericentromeric regions while the smallest were more frequently observed on chromosome extremities (Figure 4). In an A genome-wide approach, using the relative size of introgressions normalized per A chromosome (%) and their relative distance from the centromeres (%), the regression analyses revealed a positive linear relationship within AnArCo (R² = 0.66) and a positive binomial relationship within AnArCnCo (R² = 0.68) hybrids (Figure 5c; Spearman rank correlation, *p* < 2.2 × 10^−^^16^). For both AnArCn and AnArCnCo hybrids, these results demonstrate that the size of introgressions increases toward the centromeres, which is in agreement with the crossover distribution along the chromosome arms: the higher rates of recombination result in smaller sizes of introgressions. However, the significant binomial regression unraveled in the AnArCnCo hybrid translates to a less continuous gradient of introgression size from the telomere to the centromere. Strikingly, 70% of the chromosome arms carried introgressions, representing about 40% of the chromosome size in this hybrid. Analyses conducted at the scale of individual A chromosomes revealed similar results (Appendix A). The only exception is the A03 chromosome for which no significant gradient for the size of introgressions was observed in both AnArCn and AnArCnCo hybrids. This might be due to the presence of the nucleolus organizer region on the short arm of A03 [54]. For this reason, this chromosome was not considered when performing the A genome-wide approach.

### 3.3. Interest in Using the Modified Recombination Pattern Oberved in AAC Allotriploid Hybrid to Reduce the Size of a QTL Present in a B. napus Pericentromeric Region

To highlight the potential of using the modified recombination landscape observed in *Brassica* AAC allotriploids for breeding purposes, we chose as an example a QTL responsible for blackleg resistance that is present in the A01 pericentromeric region [41], at 11.8–26.7 Mb on the *B. napus* cv. Darmor-bzh v10 reference genome. The comparison of the recombination profiles from backcrossed progenies of allotriploid and allotetraploid hybrids revealed that significantly more crossovers were formed during the meiosis of the AnArCn hybrid compared to the AnArCnCo hybrid in this particular region, with 131 vs. 24 crossovers (corresponding to 58 and 16 different haplotypes), respectively (chi-square test, *p* = 1.75 × 10^−32^). At least one recombination event was detected in 70.23% vs. 11.27% of the AnArCn and AnArCnCo progenies, respectively, highlighting the highest diversity of gametes produced by the AnArCn hybrid compared to the AnArCnCo hybrid. It allowed us to break down the QTL into five regions of 0.02 to 10.7 Mb in the AnArCnCo population, and into 13 regions of 0.004 to only 4.96 Mb in the AnArCn population (Figure 6). It is important to note that the 4.96 Mb region that appeared to be deprived of crossovers in the AnArCn hybrid corresponds to a region lacking polymorphic markers, preventing the detection of the putative presence of crossovers in this region. Altogether, these results support the high potential of AAC allotriploids to reduce QTL confidence intervals.

## 4. Discussion

The production of *Brassica* AAC allotriploid hybrids, derived from the direct cross between *B. napus* and its diploid *B. rapa* progenitor, offers a unique opportunity to introgress into *B. napus* small regions of interest derived from *B. rapa* but also to highly shuffle *B. napus* diversity and decrease the size of interesting *B. napus* QTL. Indeed, this allotriploid structure allows the modification of recombination control, with an increased crossover frequency (×3.7) compared to allotetraploids, and, more interestingly, with the formation of crossovers all along the A chromosomes even in usually cold regions such as pericentromeres. These results are in agreement with previous data [36] comparing diploid and allotriploid hybrids. However, our results differ from those obtained by Leflon et al. [35], who predicted a two-fold increase in the recombination frequency in allotetraploid compared to diploid hybrids, using 18 markers of the A07 chromosome without physical anchoring on the A07 chromosome. These different results may be partly explained by the difference in marker density between the two studies, as well as by the use of different genetic backgrounds. Indeed, Pelé et al. [36] observed a slight difference (×1.4) in the recombination rates depending on the origin of the AAC allotriploid hybrid. Nevertheless, both studies agreed with the fact that AAC allotriploidy and not allopolyploidy *per se* can deeply modify the homologous recombination frequency and distribution.

### 4.1. Comparison of Breeding Strategies

Two breeding strategies can be used to introduce into oilseed rape genetic diversity or interesting traits derived from one of its diploid progenitors, *B. rapa*: either direct crosses between *B. napus* and *B. rapa* or a bridge with a resynthesized oilseed rape. For the latter, it has been clearly demonstrated that homoeologous pairing occurs during the first meiosis of synthetics [43,55,56]. Such homoeologous rearrangements between A and C chromosomes are at the origin of genomic instability with univalent and multivalent formation, resulting in reduced fertility [57,58,59,60]. Despite this genomic instability, it has been shown in several instances that homoeologous exchanges were used (sometimes inadvertently) and useful for *B. napus* breeding (for a review, see [61]). Indeed, recent sequencing data revealed that the absence of glucosinolate in seeds, a highly selected trait in oilseed rape, is partially due to a non-reciprocal translocation between A and C genomes [62]. Similarly, other translocations between A and C genomes were shown to impact several important agronomic traits, such as flowering precocity [63], disease resistance [64] or even seed quality [65]. The major difficulty in the case of exchanges between A and C genomes is to then restore the meiotic stability by decreasing the size of the introgressed region inserted in one genome to the other one. Major traits present in *B. napus* progenitors were not only introduced from resynthesized tetraploids via homoeologous recombination but also via homologous recombination. This was notably the case when introducing a clubroot resistance locus present in a *B. rapa* accession into *B.*
*napus*, leading to the *B. napus* variety Mendel, which is resistant to this disease [66,67]. However, results from the present study demonstrate that homologous recombination remains limited using resynthesized *B. napus,* making it difficult to restrict the introgression to the locus of interest. Large introgression sizes arising from this strategy, as we evidenced here, might bring undesirable loci that could reduce the agronomical value of *B. napus* elite varieties.

On the contrary, we showed that direct crosses between *B. napus* and *B. rapa* significantly improve the genetic shuffling of diversity all along the A genome. This strategy has also been largely used to generate new oilseed rape lines by backcrossing and/or selfing AAC hybrids [68,69], producing new lines with interesting heterosis. For example, the Chinese elite variety and reference genome Zs11 was obtained through this strategy [70]. The frequent use of *B. rapa* in Asian *B. napus* breeding programs may partially explain why Asian oilseed rape varieties appear as a specific group of oilseed rape varieties and why it is still possible to detect *B. rapa* introgression in this material [71]. Similarly, direct crosses between *B. napus* and *B. rapa* have been performed to introduce in *B. napus* major resistance genes to clubroot [72] or blackleg [73].

### 4.2. Optimization of Breeding Strategies

We showed that the occurrence of crossovers all along the A chromosomes, and especially in the normally cold pericentromeric regions, offers opportunities to introduce a small region of interest from *B. rapa* but also to reduce the *B. napus* QTL size. Tourrette et al. [40] found, using the data we previously obtained in *Brassica* allotriploids [36], that it is possible to decrease by a factor 10 the linkage drag in the cold region in BC3S1 after foreground selection for three backcrosses and background selection at the last selfing generation. This strategy seems, however, less efficient if the region of interest is located in a hot recombining region. Using an example of a large QTL for blackleg resistance [41] present in pericentromeric region of the A01 chromosome, we showed that the recombination rate in AAC hybrids may significantly reduce the QTL size. This result suggests that the simulation [40] can be transposed to oilseed rape breeding and can be highly efficient. The AAC hybrids offer the possibility to limit the introduction from *B. rapa* to the region of interest as well as to favor the cloning of QTL carried by *B. napus* in a cold recombinant region by breaking down the linkage disequilibrium. Thus, it will be of particular interest to more frequently apply this strategy in order to rapidly and more efficiently identify the genes underlying the different QTL identified for disease resistance or other important traits for *B. napus* (for a review, see [74]). Taking advantage of the high quality of recent *B. napus* genome assemblies [46,75], it becomes possible to define new markers, especially in pericentromeric regions, allowing a better assessment of recombination in these cold regions.

### 4.3. Development of the Strategy on Other Models

Whether the reshaping of homologous recombination observed in AAC allotriploids similarly occurs in CCA allotriploids, resulting from the cross between *B. napus* and its other diploid progenitor *B. oleracea*, remains to be deciphered (currently in progress). Even if these later CCA hybrids are more difficult to generate [76], improved recombination between homologous C chromosomes would also strongly benefit *B. napus* breeding programs. In fact, different studies revealed that it could be useful to introduce into *B. napus* new diversity from *B. oleracea* [77], such as clubroot resistance traits [78], and conversely from *B. napus* to *B. oleracea* [79]. It will also be interesting to determine if this modified homologous recombination landscape is also present in the other possible allotriploids from the U triangle [80], e.g., either AAB (deriving from B. juncea × *B. rapa*) or CCB (*B. carinata* × *B. oleracea*) hybrids, and thus possibly be useful for *Brassica* breeding.

## 5. Conclusions

Meiotic recombination is a key process that generates new genetic diversity and enables the combination of favorable alleles. However, it is strictly regulated, both in frequency and distribution. In this study, we were able to demonstrate that an AAC allotriploid hybrid presents a higher recombination rate and modified distribution compared to AA or AACC hybrids (with the latter hybrids presenting a similar recombination landscape). The boost of recombination observed in AAC hybrids allows the introduction of more and smaller genomic regions from *B. rapa* to *B. napus* compared to AACC hybrids, with a decreasing gradient of the introgression size from the centromere to the telomere in both hybrids. For breeders, the introgression of smaller genomic regions highlights the interest in using AAC allotriploid hybrids to further break the linkage disequilibrium. Additionally, the unique recombination landscape observed in AAC allotriploid hybrids will facilitate the identification of candidate genes underlying QTL of agronomical interests, most particularly in the normally cold pericentromeric regions.

## Figures and Tables

**Figure 1 biology-10-00771-f001:**
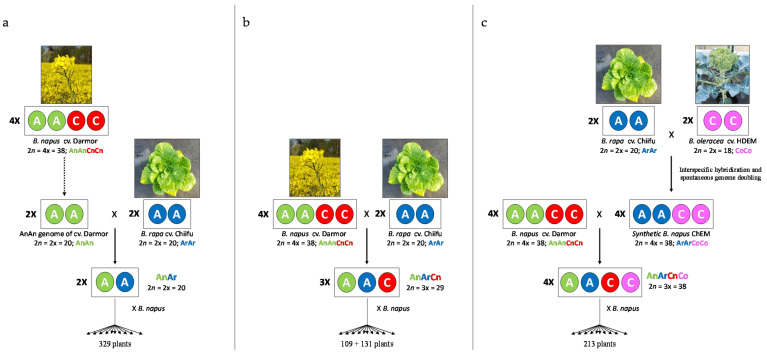
Production of (**a**) AnAr diploid, (**b**) AnArCn allotriploid and (**c**) AnArCnCo allotetraploid hybrids sharing the same A genotype and their derived progenies.

**Figure 2 biology-10-00771-f002:**
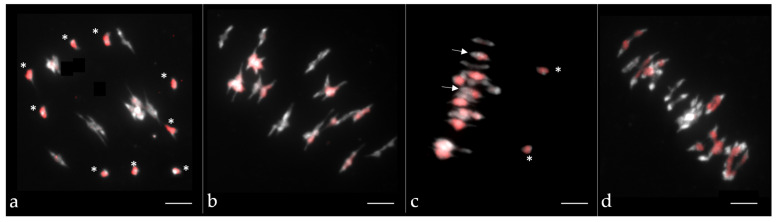
Chromosome pairing at metaphase I in pollen mother cells of (**a**) the allotriploid AnArCn, (**b**,**c**) the synthetic *B. napus* ChEM ArArCoCo-S0 and (**d**) the allotetraploid AnArCnCo hybrid. C chromosomes are labeled in red via the use of the Bob014O06 BAC clone (GISH-like). Univalents are indicated by a * and A–C bivalents by an arrow. Chromosomes were counterstained with DAPI. Scale bars represent 5 μm.

**Figure 3 biology-10-00771-f003:**
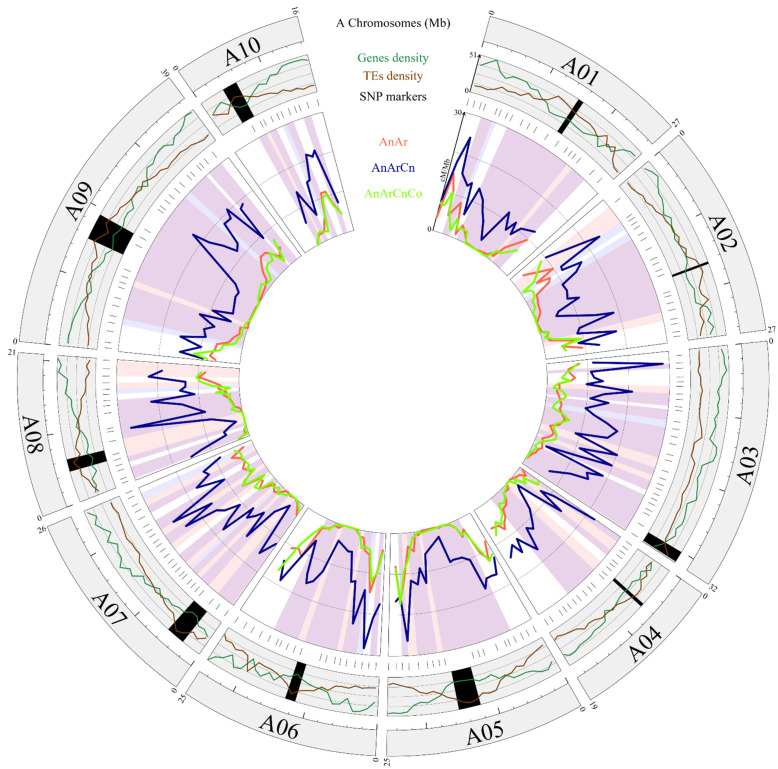
Homologous recombination landscape in diploid, allotriploid and allotetraploid hybrids. The first circle represents the 10 chromosomes of *B. rapa* cv. Chiifu v1.5. Within the second circle, the gene and transposable element (TE) densities along each A chromosome are mentioned, with black rectangles indicating the putative centromere positions [53]. In the inner circle, the recombination rates (cM/Mb) of AnAr (2*x*), AnArCn (3*x*) and AnArCnCo (4*x*) are shown as red, blue and green lines, respectively. Significant differences within an interval are highlighted in peach, light blue and purple for 2*x* vs. 3*x*, 4*x* vs. 3*x*, or 4*x* and 2*x* vs. 3*x*, respectively. Above the recombination landscape, the physical position of each polymorph SNP used for these comparisons is indicated.

**Figure 4 biology-10-00771-f004:**
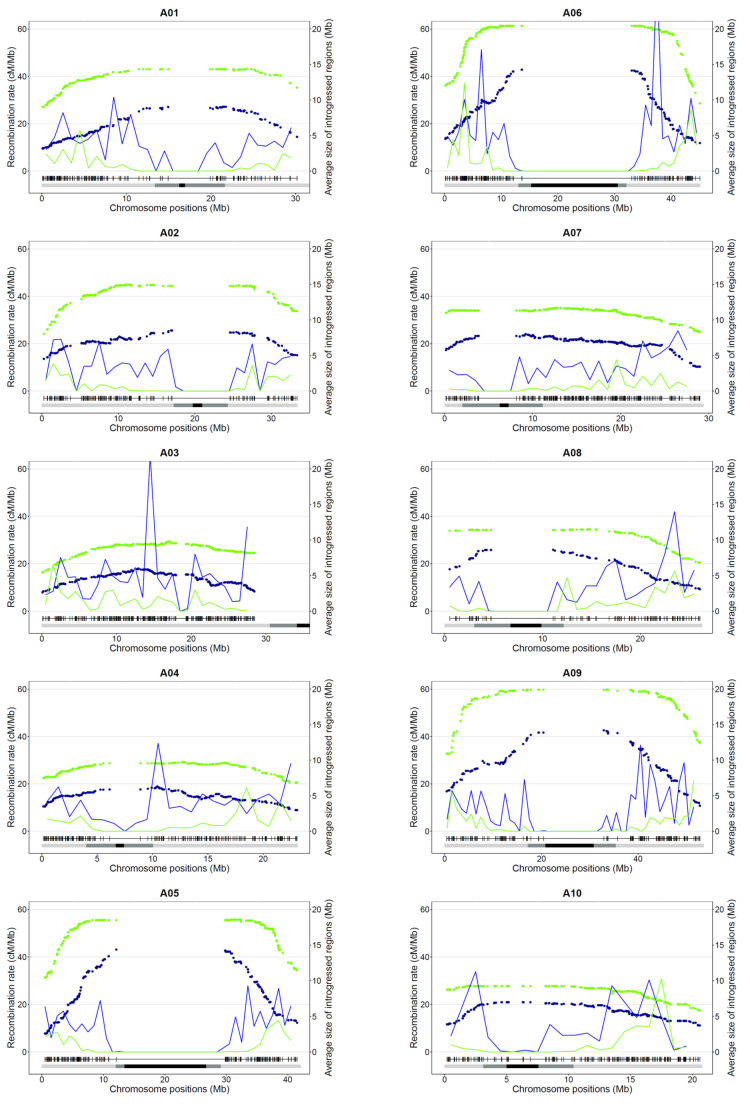
Recombination and introgression patterns along the ten A chromosomes of the allotriploid (AnArCn) and allotetraploid (AnArCnCo) hybrids. For each A chromosome (A01 to A10), the distribution of the recombination (in cM/Mb, indicated by lines) and average size of introgressions (in Mb, represented by dots) at each marker position in the progeny of allotriploid (blue) and allotetraploid hybrid (green) are presented for all A chromosomes. Below each graph, the physical positions of polymorph SNPs along each *B. napus* cv. Darmor-bzh v10 chromosome are indicated. The position of pericentromeres and centromeres (inferred in this study) are illustrated as light gray and black boxes, respectively.

**Figure 5 biology-10-00771-f005:**
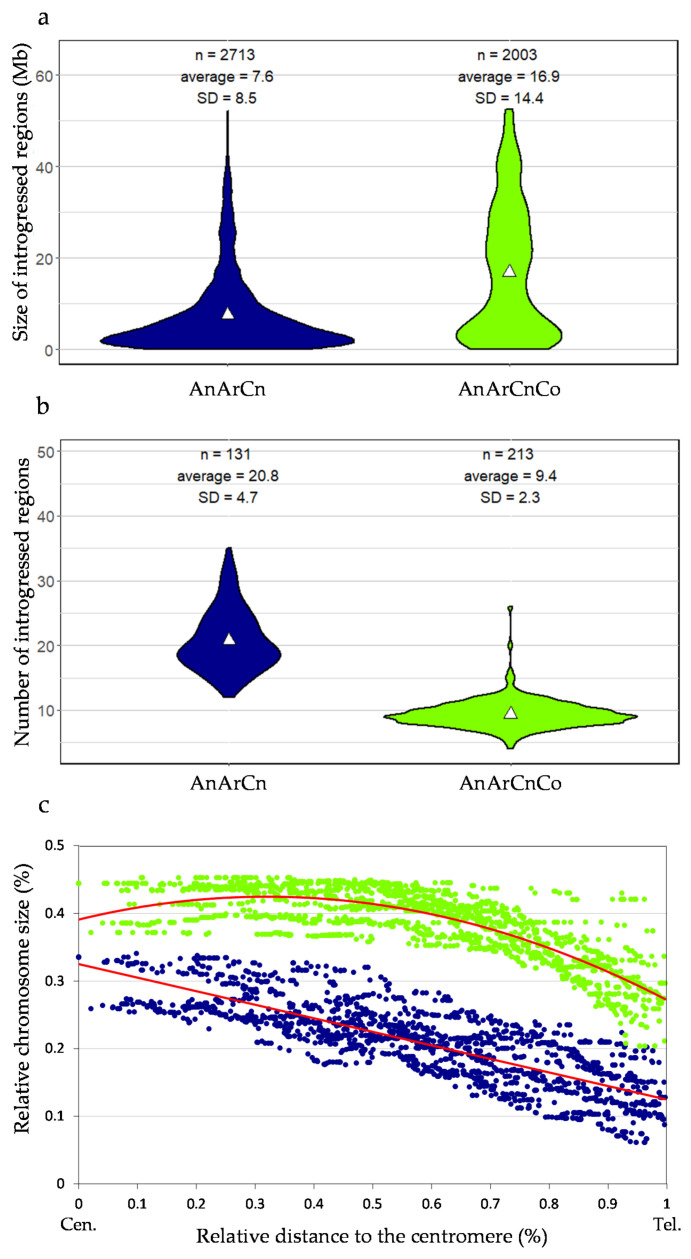
Characterization of introgressions in the AnArCn and AnArCnCo populations. (**a**) Size of introgressions, (**b**) number of introgressions. White triangle indicates mean. (**c**) Correlation between the relative introgression size and their relative position from the centromere (Cen.) to the telomere (Tel.) analyzed in the progeny of the AnArCn hybrid (blue) and AnArCnCo hybrid (green). Regression lines are indicated in red.

**Figure 6 biology-10-00771-f006:**
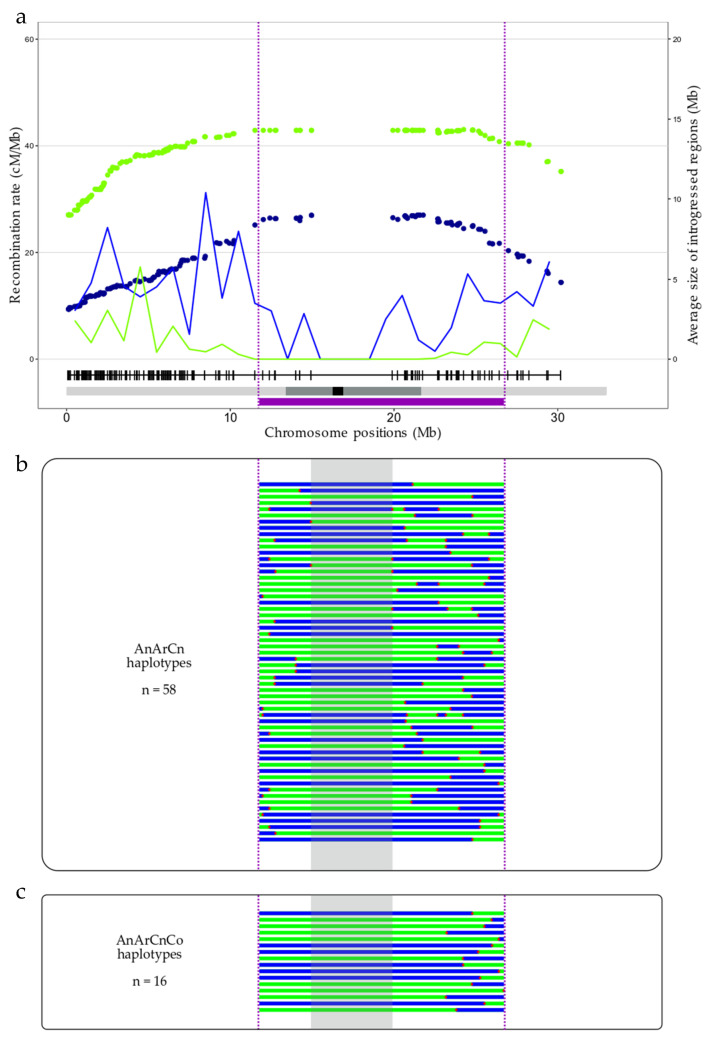
Crossover localization in a pericentromeric region carrying a QTL conferring resistance to *Leptosphaeria maculans*. (**a**) Recombination rate (in cM/Mb, as lines) and the average size of introgression (in Mb, as dots) along the A01 chromosome, for the AnArCn (green) and AnArCnCo (blue) hybrids. The resistance QTL is represented by a purple box with its borders delimited by purple dashed lines. The different possible haplotypes found in the backcross progeny of (**b**) the AnArCn (131 plants) or (**c**) the AnArCnCo hybrids (213 plants) are displayed, with the homozygous *B. napus* cv. Darmor regions in green and the introgressed regions of *B. rapa* cv. Chiifu in blue. A 4.96 Mb region surrounding the centromere and deprived of markers is highlighted in gray. Red stars symbolize the crossover positions.

## Data Availability

Genotyping data of the 202 SNP used for the AA and AAC hybrids were retrieved from Pelé et al. (2017) (available online: https://journals.plos.org/plosgenetics/article?id=10.1371/journal.pgen.1006794#sec024, accessed on February 2021).

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
