# Peer review of "A Modified Meiotic Recombination in Brassica napus Largely Improves Its Breeding Efficiency"

_biology, 2021, doi:10.3390/biology10080771_

Round 1
Reviewer 1 Report
Franz Boideau and colleagues described increased meiotic recombination in Brassica allotriploid AAC hybrid, and its utilization in improving Brassica napus breeding efficiency. The authors provided extensive evidences based on carefully designed combinations of AnAr, AnArCn and AnArCnCo. The modified meiotic recombination in allotriploid AAC hybrid has indeed potential to improve Brassica napus breeding efficiency. I would recognize it as an useful breakthrough in Brassica breeding. However, I still have some major concerns to be addressed.
- I hope the authors would prove or at list discuss if the allotriploid but not the specific allotriploid AAC hybrid would increase the recombination. For example, will allotriploid ACC hybrid increase the recombination of the C subgenome?
- Will different C or A genotype influence the meiotic recombination of the allotriploid?
Author Response
Reviewer 1 comments:
Franz Boideau and colleagues described increased meiotic recombination in Brassica allotriploid AAC hybrid, and its utilization in improving Brassica napus breeding efficiency. The authors provided extensive evidences based on carefully designed combinations of AnAr, AnArCn and AnArCnCo. The modified meiotic recombination in allotriploid AAC hybrid has indeed potential to improve Brassica napus breeding efficiency. I would recognize it as an useful breakthrough in Brassica breeding. However, I still have some major concerns to be addressed.
- I hope the authors would prove or at list discuss if the allotriploid but not the specific allotriploid AAC hybrid would increase the recombination. For example, will allotriploid ACC hybrid increase the recombination of the C subgenome?
- Will different C or A genotype influence the meiotic recombination of the allotriploid?
Responses:
- To date, there is no studies that tested if other Brassica allotriploids (such as ACC hybrids) may present this modified homologous recombination landscape observed in AAC hybrids; Such work is currently in progress in our lab and shall soon be published.
To clarify this important point, we added the following paragraph within the revised version of the manuscript (l 481 - 491).
‘Whether the reshaping of homologous recombination observed in AAC allotriploids similarly occur in CCA allotriploids, resulting from the cross between B. napus and its other diploid progenitor B. oleracea, remains to be deciphered (currently in progress). Even if these later CCA hybrids are more difficult to generate [76], improved recombination between homologous C chromosomes would also strongly benefit B. napus breeding programs. In fact, different studies revealed that it could be useful to introduce into B. napus new diversity from B. oleracea [77], such as clubroot resistance traits [78], and conversely from B. napus to B. oleracea [79]. It will also be interesting to determine if this modified homologous recombination landscape is also present for the other possible allotriploids from the U triangle [80] , e.g. either AAB (deriving from B. juncea x B. rapa) or CCB (B. carinata x B. oleracea) hybrids and thus maybe useful for Brassica breeding.'
We modulated our manuscript by adding “AAC allotriploidy” line 125, line 392 and line 407.
- Related to the putative genotype effect on homologous recombination rate in different AAC hybrids, we added the following sentence (l 403-409):
‘These different results may be partly explained by the difference of marker density between the two studies, as well as by the use of different genotic background. Indeed, Pelé et al. [36] observed a slight difference (x1.4) of the recombination rates depending on the origin of the AAC allotriploid hybrid. Nevertheless, both studies agreed with the fact that AAC allotriploidy and not allopolyploidy per se can deeply modify the homologous recombination frequency and distribution.’
Reviewer 2 Report
Dear Authors,
I revised the manuscript number biology-1317319 titled: “ A modified meiotic recombination in Brassica napus largely improves its breeding efficiency”.
The experimental design, the results as well as the discussion are congruent with the objectives and well explained. For these reasons, to my opinion, the manuscript is publishable in Biolgy after a few minor remarks attached in the file.
Best regards

Author Response
Comments and responses to reviewer 2:
Line 21: occur along the entire A
We agreed and modified this.
Line 27-28 are almost the same 46-48. Please change them and try to explain better the issue.
We modified the sentence l 47-49 to prevent redundancy as following:
‘Meiotic recombination shuffles parental alleles to produce new allelic combinations in the progenies, hence producing new genetic diversity at each generation. This biological mechanism is a key evolutionary process that is commonly used in plant breeding to accumulate favorable alleles and purge deleterious mutations [1-3].’
Line 67: Check with the reference style of the journal
We corrected this and updated the references.
Line 390: replace « maybe » by « are »
We preferred to keep the « maybe » but added the following sentence: « These different results may be partly explained by marker density, as well as by the use of different genotype background. Indeed, Pelé et al. (2017) observed a slight difference (x1.4) of the recombination rates depending on the origin of the AAC allotriploid hybrid. »
Reviewer 3 Report
This manuscript demonstrates crossover rate was higher with smaller and numerous introgressions of B. rapa in AAC hybrids compared to AA and AACC hybrids. It looks very interesting and good overall. Please check the following:
- Line 23 theseallotriploid: Please add space
- Line 184 AnAr (, the AnArCn: Please delete (,
- Please check acronyms CO in line 225 and NOR in line 345
Author Response
Comments and responses to reviewer 3:
Please check the following:
- Line 23 theseallotriploid: Please add space
- Line 184 AnAr (, the AnArCn: Please delete (,
- Please check acronyms CO in line 225 and NOR in line 345
We thank the reviewer 3 for these comments and corrected the two typographical errors and clarified the two acronyms.
Reviewer 4 Report
Control of meiotic recombination is one of the main tools to especially increase the genetic diversity under genetic erosion or to introduce small genomic regions containing cold QTLs. In this paper, authors showed that it is possible to modify the meiotic crossing overs under tight control in oilseed rape. Authors performed various molecular genetic analyses for three genetic hybrids between B. napus and B. rapa with different ploidy levels, AA, AAC and AACC, which share genetically identical A genome sequence. The analyses were resulted in 3.7-fold increase of crossover rate and smaller and numerous introgressions of B. rapa regions in the allotriploid AAC as compared to AA and AACC hybrids. Authors further showed that the meiotic recombination boost observed in the allotriploid can be applied to reduce the size of QTL resistant to blackleg, which is present in a cold pericentromeric region in Brassica. This paper has novelty and is worth to be published in the journal Biology. This paper will be read by plant biologists and crop breeding scientists etc.
Minor comments,
In Figure 5, according to the text, Figs. 5a and 5b should be switched each other.
Several typographical errors to be amended,
Line 23, theseallotriploid
Line 429, progenitorswere
Author Response
Comments and responses to reviewer 4:
Minor comments,
In Figure 5, according to the text, Figs. 5a and 5b should be switched each other.
We switched the two figures 5a and 5b to match the text and legend.
Several typographical errors to be amended,
Line 23, theseallotriploid
Line 429, progenitorswere
We added spaces for these two typographical errors.